# An Experimental Investigation into the Feasibility of a DC Hybrid Power Plant for a Northern Sea Route Ship

Yi Zhou [1], Kayvan Pazouki [1,*], Rose Norman [1], Haibo Gao [2] and Zhiguo Lin [2]

1   Marine, Offshore and Subsea Technology Group, School of Engineering, Newcastle University, Newcastle upon Tyne NE1 7RU, UK; yi.zhou2@newcastle.ac.uk (Y.Z.); rose.norman@newcastle.ac.uk (R.N.)
2   School of Energy and Power Engineering, Wuhan University of Technology, Wuhan 430063, China; hbgao_whut@126.com (H.G.); zglin@139.com (Z.L.)
*   Correspondence: kayvan.pazouki@newcastle.ac.uk

**Abstract:** Increasingly, the melting of Arctic ice due to global warming has provided opportunities for commercial shipping between Asia and Europe. Given the vulnerability of the Arctic environment, especially due to emissions of short-lived pollutants from shipping activities, a more effective propulsion system with a comprehensive control strategy is required to reduce fuel consumption, thus potentially mitigating the impacts of shipping activities on the northern sea route (NSR). In this paper, a shipboard DC hybrid system powered by a combination of diesel generator sets and batteries is proposed and analysed in terms of its application on a ship in the NSR. The specific fuel consumption and various losses in the power sources were analysed to develop an efficiency-optimisation control strategy for the proposed DC hybrid power system. To evaluate the performance of the hybrid power system with the proposed optimisation control strategy, lab-scale experiments have been conducted in the Shanghai Marine Diesel Engine Research Institute to compare the proposed system with a conventional hybrid system. The experimental results indicate that the proposed DC hybrid power plant with the energy optimisation control contributes a 5.35% fuel saving compared with the DC fixed-speed diesel electric configuration during a scaled-down NSR scenario.

**Keywords:** hybrid power system; modelling; power management; energy management; northern sea route shipping

## 1. Introduction

The northern sea route (NSR) is considered to be a part of the shortest route between northeast Asia and northern Europe [1]. Although global warming has caused a reduction in the Arctic Ocean's ice at the rate of 13% every decade [2], this offers new opportunities for the exploration of polar regions. However, the associated environmental impacts of exploiting the shortest route need to be investigated. Several studies have analysed the impacts of international shipping on climate and air pollution [3–6], and these have indicated that ships contribute to global climate change and health impacts. To relieve the deteriorating situation, the International Maritime Organization (IMO) has proposed stringent regulations to reduce emissions and enhance the energy-efficiency of the shipping industry [7]. At present, shipping in the NSR only contributes a relatively small proportion of global shipping emissions. However, regional effects from emissions, such as black carbon (BC) and ozone ($O_3$), must be understood and reduced in the near future [8].

The use of energy-efficient and low-emission technologies in icebreaker propulsion systems has been a subject of investigation since the 1930s. The first diesel–electric icebreaker ship was built in 1933 [9] and had improved low speed torque characteristics when compared with traditional direct mechanical systems. In 1957, the first nuclear-powered icebreaker ship was commissioned by the Soviet Union [10], which made it possible to

explore the Arctic area in extreme environmental conditions. However, these improvements came with new challenges. Under heavy load conditions, the need for redundancy means that there are engines in diesel electric systems running at low part load, which results in poor fuel consumption and increased emissions [11]. Although variable-speed operation has been developed and utilised in DC diesel electric configurations [12], the optimisation range is limited in speed regulation, and inefficiently running engines still underperform at some specific loading conditions. Moreover, Hodge and Mattick [13] proposed that variable-speed operation might not be suitable for ice-capable ships from the perspective of environmental concerns in fragile Arctic areas, as variable-speed operation tends to increase the emission of $NO_2$. In terms of nuclear technology, any leakage of radioactive materials into the sensitive Polar region would cause detrimental damage to Arctic ecosystems [14].

Given the limitations of and concerns with the traditional propulsion systems, hybrid propulsion technology with energy storage systems (ESS) could provide a suitable option in terms of fuel-efficiency and environmental considerations. Due to the limitations of low energy density, it would be still challenging for most types of ships to have all-electric propulsion systems supplied from an ESS [15]. Therefore, hybrid power systems may be preferable, as the low energy density of the ESS can be compensated by conventional diesel engines. Benefits, such as increased energy efficiency, flexibility, and reliability, can be achieved by using hybrid systems when compared with conventional configurations [16]. Most notably, hybrid propulsion can reduce fuel consumption and emissions by optimizing the operations of the power sources (diesel generator and ESS) over a wide range of load power [17].

According to Geertsma [11], by applying hybrid architecture with advanced control strategies, fuel consumption and emission reduction can reach up to 35%. In the history of hybrid technology, the AC hybrid power system has been dominant because of the well-developed control, safety, and voltage transformation systems [18]. However, it has drawbacks in the form of harmonics, synchronization requirements, and fixed speed operation. In contrast, the DC power system reduces the complexities of the AC system, but the development of comprehensive control strategies has lagged behind. With the development of circuit protection and advanced control strategies in recent years, the utilization of DC distribution systems has been increasingly prevalent in hybrid power generation [19]. Companies such as ABB and Siemens have developed hybrid DC distribution in ship power systems [11]. Moreover, in the field of battle ships, the US Navy has also developed a DC distribution system for its DDG-1000 destroyers [20].

Several advanced control strategies for ship DC distribution hybrid systems have been developed. To consider the uncertainty and demand of hybrid power systems, Haseltalab et al. [21] and Park et al. [22] proposed a Model Predictive Control (MPC) method for tackling conflicting requirements, as well as predicting future loads. However, although the MPC technique can relieve the negative impact of future error, it has been shown to be sensitive to parameter variations. In addition, filters are required to deal with the computational burden, which increases the system complexity [23]. In terms of higher-level control strategies, Al-Falahi et al. [24] developed a hybrid power management control strategy for hybrid electric ferries. While the significance of optimizing the efficiency of the energy storage system, as well as the optimal selection of the engine's Specific Fuel Consumption (SFC) operating region, has been presented in their study, they did not consider the efficiency in relation to the output power of the diesel generator, which limits the scope of their investigation. Yuan et al. [25] and Bui et al. [26] proposed energy management strategies for hybrid power systems in ships. Although the performance of the target hybrid system was enhanced in terms of fuel consumption reduction, the power losses in the diesel generator and power converters were not considered in either of their studies, which might cause a significant shift in the optimal operating points of power sources under heavy loading conditions. Although losses in power sources have been considered in Jianyun et al. [27] and Bui et al. [26], the variation in efficiency coefficients under variable loading conditions needs to be further addressed.

In this paper, a holistic dynamic model of the DC marine hybrid power system is developed. Both the low- and high-level control systems required to regulate various control objectives at the component and system levels are developed. In addition, an Efficiency Optimisation Algorithm is developed to determine the optimal modes of the proposed hybrid system and the shifted optimal power set-points due to losses in the power sources. The proposed model is validated with experimental results from a scaled-down NSR scenario. In order to demonstrate the performance of a DC shipboard hybrid system and the optimisation algorithm, fixed-speed and variable-speed DC diesel electric systems, a hybrid system with conventional three-level control, and a hybrid system with the proposed efficiency optimisation algorithm have all been modelled and evaluated experimentally in terms of system stability and fuel consumption.

This paper is structured as follows. Section 2 provides a comprehensive review and analysis of the mathematical models pertaining to the system components. The proposed control strategy and optimisation algorithm are discussed in Section 3. Section 4 then presents the simulation and experimental setups, along with the results obtained for simulation validation. Furthermore, Section 5 undertakes a comparative analysis of different system configurations under diverse control strategies, based on the experimental findings. Finally, Section 6 presents the conclusions drawn from the work.

## 2. Modelling of the DC Hybrid Power System

A simplified block diagram of the hybrid power system modelled in this paper is shown in Figure 1. The two diesel engines drive their respective generator sets, which are interfaced to the DC bus through rectifiers. A bidirectional DC–DC converter connects the battery bank to the bus. These power sources work together to supply power to the propulsion motors. Overall, the power sources have been connected together using the DC bus, the rectifiers, and the converters, which have been modelled and implemented in MATLAB SIMULINK according to the following mathematical modelling procedures.

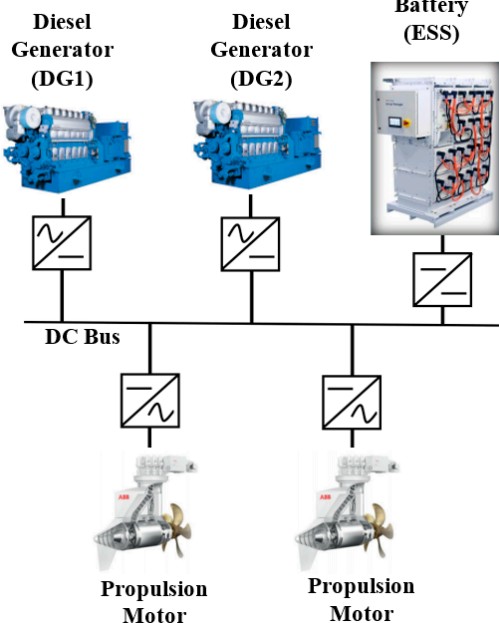

**Figure 1.** Typical configuration of shipboard hybrid DC power systems.

### 2.1. Generator Sets

Firstly, a simplified mathematical model of a diesel engine is presented. The transfer function is expressed as follows:

$$T_m(s) = u_c(s) \left( \frac{K_a}{T_1 s + 1} \right) \left( \frac{1}{t_d s + 1} \right) \tag{1}$$



where $u_c(s)$ is a control signal from the controller, which is based on the speed error, $K_a$ is the actuator gain and $T_1$ is the time constant of the actuator, $T_m$ is the torque produced by the engine, and $t_d$ is a time-delay constant. DG1 is a synchronous generator. The full-order synchronous machine model is given as follows [28]:

$$\dot{\boldsymbol{\psi}}_s = \omega_b \boldsymbol{U}_s + \omega_b \boldsymbol{R}_s \boldsymbol{I}_s + \omega_{sr} \boldsymbol{\psi}_s \tag{2}$$

$$\dot{\boldsymbol{\psi}}_{kf} = \omega_b \left( \boldsymbol{E}_f - \boldsymbol{R}_{kf} \boldsymbol{I}_{kf} \right) \tag{3}$$

where $\boldsymbol{\psi}_s = \left[ \psi_d, \psi_q, 0 \right]^T$ and $\boldsymbol{\psi}_{kf} = \left[ \psi_{kd}, \psi_{kq}, \psi_f \right]^T$ are the DG1 magnetic flux vector, $\boldsymbol{I}_s = \left[ i_{ds}, i_{qs}, 0 \right]^T$ and $\boldsymbol{I}_{kf} = \left[ i_{kd}, i_{kq}, i_f \right]^T$ are the current vector, $\boldsymbol{U}_s = \left[ u_{ds}, u_{qs}, 0 \right]^T$ is the voltage vector, $\boldsymbol{E}_f = \left[ 0, 0, v_f \right]^T$ is field voltage vector, $\boldsymbol{R}_s = \begin{bmatrix} -r_s & 0 & 0 \\ 0 & -r_s & 0 \\ 0 & 0 & 0 \end{bmatrix}$ and

$\boldsymbol{R}_{kf} = \begin{bmatrix} r_{kd} & 0 & 0 \\ 0 & r_{kq} & 0 \\ 0 & 0 & r_f \end{bmatrix}$ are resistance matrices, and $\omega_{sr} = \begin{bmatrix} \omega_{sr} & 0 & 0 \\ 0 & -\omega_{sr} & 0 \\ 0 & 0 & 0 \end{bmatrix}$ is the speed matrix.

$$\boldsymbol{I}_s = \boldsymbol{X}_s \left[ \boldsymbol{\psi}_s - \boldsymbol{u} \boldsymbol{\psi}_m \right] \tag{4}$$

$$\boldsymbol{I}_{kf} = \boldsymbol{X}_{kf} \left( \boldsymbol{\psi}_{kf} - \boldsymbol{\psi}_m \right) \tag{5}$$

$$\boldsymbol{u}\boldsymbol{\psi}_m = x_{MQ} \left( u_2 \boldsymbol{X}_{kf} u_1 + \boldsymbol{X}_s u_3 \right) \left( u_2 \boldsymbol{\psi}_s + u_1 \boldsymbol{\psi}_{kf} \right) + x_{MD} \left( \boldsymbol{X}_s u_{1,4} + u_6 \boldsymbol{X}_{kf} u_{5,7} + u_4 \boldsymbol{X}_{kf} u_{3,8} \right) \left( u_3 \boldsymbol{\psi}_s + u_9 \boldsymbol{\psi}_{kf} \right) \tag{6}$$

where $\boldsymbol{X}_s = \begin{bmatrix} x_s & 0 & 0 \\ 0 & x_s & 0 \\ 0 & 0 & 0 \end{bmatrix}^{-1}$ and $\boldsymbol{X}_{kf} = \begin{bmatrix} x_{kd} & 0 & 0 \\ 0 & x_{kq} & 0 \\ 0 & 0 & x_f \end{bmatrix}^{-1}$ are leakage reactance matrices,

$\boldsymbol{\psi}_m = \left[ \psi_{md}, \psi_{mq}, \psi_{md} \right]^T$ is the magnetizing flux vector, and $\boldsymbol{u}_i$ is the permutation matrix group. $x_{MD}$ and $x_{MQ}$ are integrated dq reactance. $\omega_b$ is base electrical angular speed.

Due to the lab setup, DG2 is an asynchronous generator with parameters similar to DG1. Although the system structure is not a usual ship electrical power system configuration, which would normally consist of two or more synchronous machines, it can offer the required performance in a DC shipboard system. The mathematical model is expressed as follows [29]:

$$\dot{\boldsymbol{\psi}}_{as} = \omega_b \boldsymbol{U}_a + \omega_b \boldsymbol{R}_{as} \boldsymbol{I}_{as} + \omega_e \boldsymbol{\psi}_s \tag{7}$$

$$\dot{\boldsymbol{\psi}}_r = \omega_b \boldsymbol{R}_r \boldsymbol{I}_r + \omega_s \boldsymbol{\psi}_r \tag{8}$$

where $\boldsymbol{\psi}_{as} = \left[ \psi_{ds}, \psi_{qs} \right]^T$ and $\boldsymbol{\psi}_r = \left[ \psi_{dr}, \psi_{qr} \right]^T$ are the DG2 magnetic flux vector, $\boldsymbol{I}_{as} = \left[ i_{da}, i_{qa} \right]^T$ and $\boldsymbol{I}_r = \left[ i_{dr}, i_{qr} \right]^T$ are the current vector, $\boldsymbol{U}_a = \left[ u_{da}, u_{qa} \right]^T$ is voltage vector, $\boldsymbol{R}_{as} = \begin{bmatrix} -r_{as} & 0 \\ 0 & -r_{as} \end{bmatrix}$ and $\boldsymbol{R}_{ar} = \begin{bmatrix} r_{ar} & 0 \\ 0 & -r_{ar} \end{bmatrix}$ are resistance matrices, and $\omega_e = \begin{bmatrix} \omega_e & 0 \\ 0 & -\omega_e \end{bmatrix}$ and $\omega_s = \begin{bmatrix} \omega_e - \omega_r & 0 \\ 0 & -(\omega_e - \omega_r) \end{bmatrix}$ are speed and slip matrices, respectively.

$$\boldsymbol{I}_{as} = \boldsymbol{X}_{as} \left( \boldsymbol{\psi}_s - \boldsymbol{\psi}_{m1} \right) \tag{9}$$

$$\boldsymbol{I}_r = \boldsymbol{X}_r \left( \boldsymbol{\psi}_r - \boldsymbol{\psi}_{m1} \right) \tag{10}$$

$$\psi_{m1} = x_M(X_{as}\psi_s + X_r\psi_r) \tag{11}$$

where $X_{as} = \begin{bmatrix} x_{ls} & 0 \\ 0 & x_{ls} \end{bmatrix}^{-1}$ and $X_r = \begin{bmatrix} x_{lr} & 0 \\ 0 & x_{lr} \end{bmatrix}^{-1}$ are leakage reactance matrices. $\psi_{m1} = \left[\psi_{md1}, \psi_{mq1}\right]^T$ is a magnetizing flux vector. $x_M$ is the integrated reactance [29].

For both of the above generators, the mechanical torque can be expressed as follows:

$$T_m - T_e = J\dot{\omega}_{rotor} + D\omega_{rotor} \tag{12}$$

$$T_e = \frac{3P}{4\omega_b}\left(i_q\psi_d - i_d\psi_q\right) \tag{13}$$

where $T_e$ is the electric torque, $\omega_{rotor}$ is the rotor speed, $J$ is the inertia of the generator, $D$ is the damping coefficient of rotor, and $P$ is the number of poles.

### 2.2. Lithium-Ion Battery

In this paper, lithium-ion battery packs are used as the ESS because of their relatively high-energy density and good dynamic response [30]. The generic battery model for the lithium-ion battery has been implemented in MATLAB Simulink for simulation model development. Figure 2 outlines its implementation in Simulink.

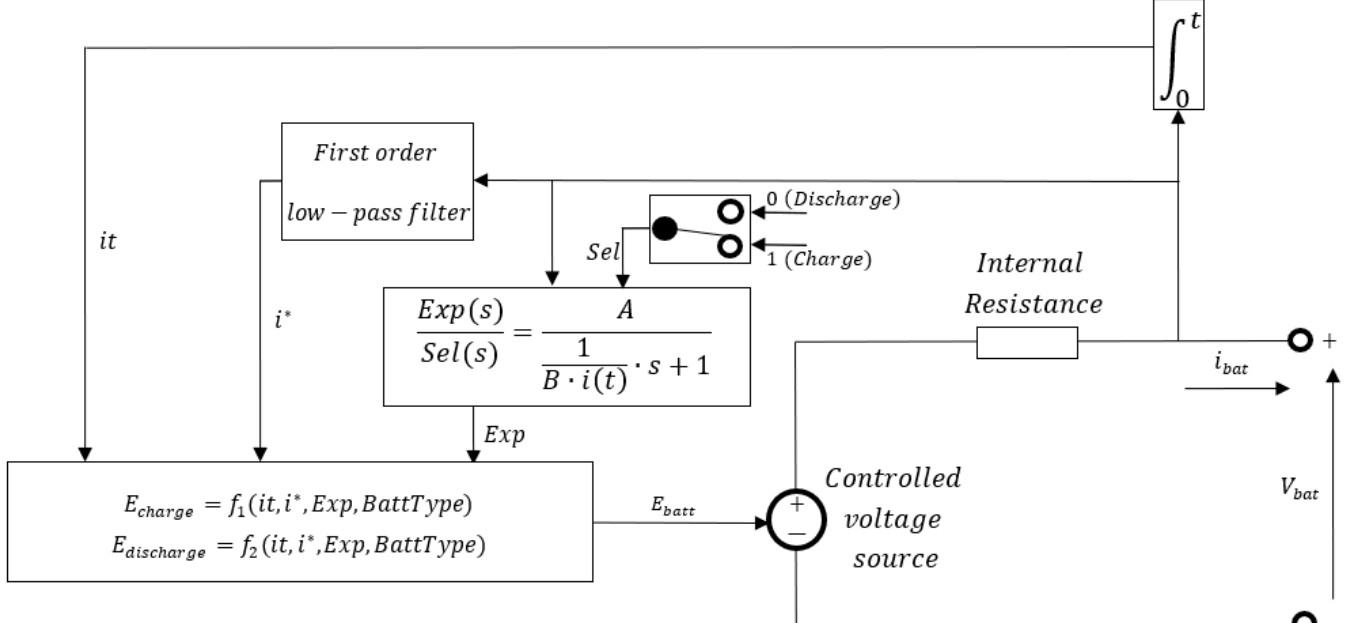

**Figure 2.** Battery generic model.

The mathematical model of a Li-ion battery can be built based on the following equations [30]:

Discharging:

$$V_{Bat} = E_0 - Ri_{bat} - K\frac{Q}{Q-it}it + Aexp(-B \times it) - K\frac{Q}{Q-it}i^* \tag{14}$$

Charging:

$$V_{Bat} = E_0 - Ri_{bat} - K\frac{Q}{Q-it}it + Aexp(-B \times it) - K\frac{Q}{it-0.1Q}i^* \tag{15}$$

where $V_{Bat}$ is battery voltage, $E_0$ is battery constant voltage, $K$ is polarisation constant, $Q$ is battery capacity, *it* is actual battery charge, calculated as $it = \int idt$, $R$ is internal resistance, $i_{bat}$ is battery current, $i^*$ is filtered current, $A$ is the exponential zone amplitude, and $B$ is the inverse of the exponential zone time constant. This battery generic model has been validated by Tremblay and Dessaint [30]. In their work, the simulation results demonstrated good agreement with the datasheet from the manufacturer. The application of a model in a hybrid power system has been introduced in the work of Evangelou and Shukla [31] and Bassam et al. [32].

### 2.3. Average-Value-Model (AVM) Rectifier

The rectifier system is modelled using the average-value model (AVM) technique in this work. The main aim of the AVM rectifier is to achieve rectifier operation through establishing a relationship between the DC-link variables on one side and the AC variables transferred to a suitable reference frame on the other side. Figure 3 shows the scheme of an AVM rectifier with the diesel generator rectifier system (DGRS) based on Jatskevich et al. [28] and Shahab [33]. In their research, several AVM parameters are applied to express the voltage ratio between the DC side and the AC side, the current ratio, and the phase shift between the fundamental harmonics of the generator voltage and current, respectively. The AVM parameters can be either constants or variables. In this work, in order to have better performance in terms of accuracy at a high load power, the AVM model parameters were considered variable as justified by Zahedi and Norum [34]. Through this strategy, the inputs and outputs of each DGRS set can be related together based on load current $i_{load}$, which means the AVM rectifier system is used to provide the dq-axis voltage as inputs for the generator models, and the generators will provide the dq-axis current for the AVM rectifier system.

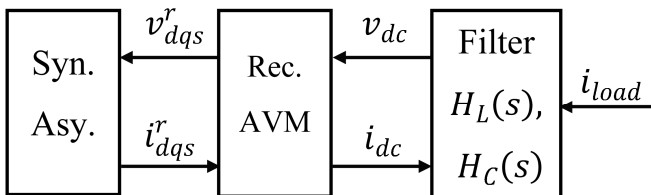

**Figure 3.** Block diagram of AVM rectifier.

The mathematical model of the Rec. AVM is presented as follows:

$$v_{dqs}^r = \alpha(z)v_{dc}\begin{bmatrix}\cos(\delta)\\\sin(\delta)\end{bmatrix} \tag{16}$$

$$\delta = \arctan\left(\frac{i_{ds}^r}{i_{qs}^r}\right) - \phi(z) \tag{17}$$

$$i_{dc} = \beta(z)\sqrt{i_{ds}^2 + i_{qs}^2} \tag{18}$$

The mathematical model of filters is expressed as follows:

$$H_L(s) = \frac{L_f s}{\tau s + 1} \tag{19}$$

$$H_C(s) = \frac{1}{C_f s} \tag{20}$$

$$v_C = H_C(s)(i_{dc} - i_{load}) \tag{21}$$

$$v_{dc} = v_C + H_L(\text{s})i_{dc} \tag{22}$$

where δ is the angle between phase A voltage and the dq reference frame, which is dependent on the load. α, β, and $\phi$ are the parameters of the AVM based on impendence $z$, which are the voltage ratio between the DC side and AC side, the current ratio, and the phase shift between the harmonics of generator voltage and current, respectively. $L_f$ and $\tau$ are the inductance and time constant of $H_L(\text{s})$. $C_f$ is the filter capacitor and $v_C$ is the capacitor voltage.

### 2.4. Bidirectional DC–DC Converter

A bidirectional DC–DC converter model was developed using a dual half-bridge converter topology with soft zero-voltage-switching (ZVS) to reduce the losses, as switching in high voltage and current conditions may lead to hard switching, which will cause losses [35]. The converter can be represented by an equivalent circuit for DC–DC conversion, as shown in Figure 4.

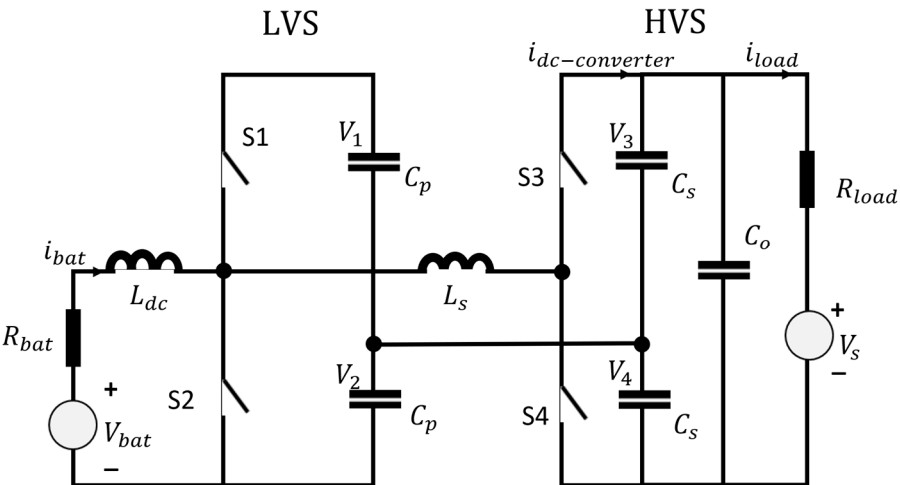

**Figure 4.** The equivalent circuit of soft-switched bi-directional half-bridge DC–DC converter.

The simplified average value model for bidirectional power flow can be expressed as follows:

$$\frac{di_{bat}}{dt} = -\frac{R_{bat}}{L_{dc}}i_{bat} - \frac{1}{2L_{dc}}V_{12} + \frac{1}{L_{dc}}V_{bat} \tag{23}$$

$$\frac{dV_{12}}{dt} = \frac{1}{C_P}i_{bat} - \frac{2\varphi(\pi - \varphi)}{2C_P T_S \omega^2 L_S}V_{34} \tag{24}$$

$$\frac{C_t}{2}\frac{dV_{34}}{dt} = i_{dc-converter} - i_{load} \tag{25}$$

$$i_{dc-converter} = \frac{\varphi(\pi - \varphi)}{2T_S \omega^2 L_S}V_{12} \tag{26}$$

$$C_t = C_s + 2C_o \tag{27}$$

where $T_S$ is the sample time, $\varphi$ is the phase shift regulated by the adjustable inductance $L_S$, $i_{dc-converter}$ is the DC current from the converter, $i_{bat}$ is the battery output current, $V_{12}$ is the average voltage on the Low Voltage Side (LVS), and $V_{34}$ is the average DC-link voltage on the High Voltage Side (HVS).

## 3. Control Strategy and Optimisation Algorithm

Three-level hierarchical control has been shown to be one of the most efficient structures for a system with multiple control aims [36]. Multiple control objectives are required to be achieved in the DC hybrid power system, such as maintaining the stability of the bus voltage, the ESS current, the engine speed, and the DGRS power, and controlling the system to operate within certain boundaries. Thus, in previous works from Ghimire et al. [18] and Chua et al. [37], three different control levels have been designed for the generation side, namely tertiary (Ter.) and secondary (Sec.) controls, aiming to improve the fuel efficiency of the system, and the primary (Pri.) control to regulate the voltage and frequency according to the reference signals. However, due to the losses in the power converters and power sources, which would vary with load change, the optimal power set-point would be shifted according to load demand. In some specific loading conditions, for example, around a full load in the diesel engine, the diesel electric mode (operation without ESS) is even more fuel efficient than the hybrid mode (operation with ESS) [11]. In this work, the conventional hierarchical control is improved by developing an efficiency optimisation algorithm to calculate the shifted optimal power set-point for further fuel saving. Figure 5 presents an overview of the proposed control strategy for a DC hybrid power system. The regulators and governors, as well as their input and output control signals, are also shown. The load side inverter drive is replaced by a controllable resistance for the purpose of emulating the target load profile. In this work, PID controllers are mainly applied in the hierarchical control strategy. The design criterion for the PID controllers applied in this research is zero steady-state error, where the integral gain in the PID is carefully tuned to make the error converge to zero, ensuring zero steady-state error. The integral control is particularly effective for processes with constant or slowly changing disturbances that would otherwise cause a steady-state error in a proportional-only control scheme.

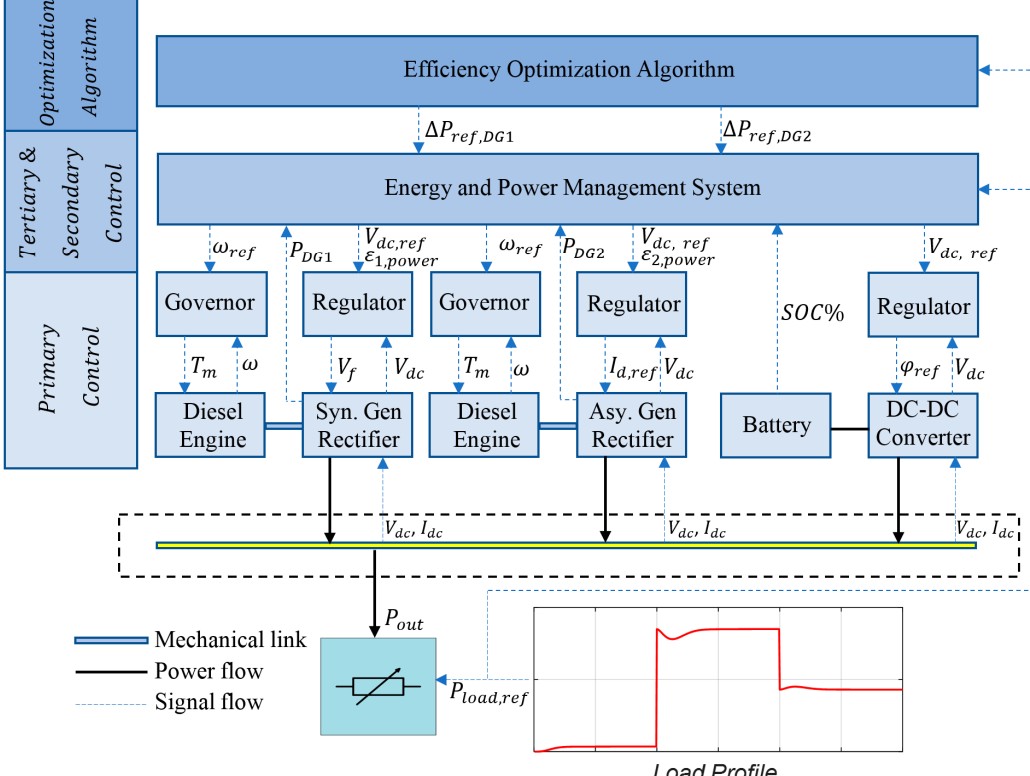

**Figure 5.** Overview of the proposed control system for a DC hybrid power system.

### 3.1. Voltage and Frequency Control

The primary control strategy in the DC hybrid power system aims to regulate the power, voltage, or speed according to the reference signals obtained from a higher-level controller.

(1) Governor: The speed of the diesel engines is regulated by PID controllers to provide reference torque to drive the diesel generator rectifier system, which is based on the speed error between the reference speed and the actual speed. The block diagram of the diesel engine speed control is presented in Figure 6, where $\omega^*$ is reference speed, $\omega$ is the measured speed normally obtained from $\omega_{rotor}$ in Equation (12), $\varepsilon$ is the speed error, and $u_c$ is the control signal for Equation (1).

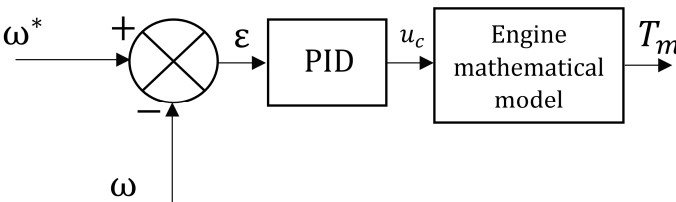

**Figure 6.** Block diagram of engine governor.

(2) Regulator of the Generator Rectifier System: The generator voltage is usually maintained using a voltage regulator. In this work, the DC bus voltage is directly regulated using PI controllers, which regulate the generator's excitation field voltage $V_f$ in the synchronous generator (DG1) and reference d-axis current $I_{d,ref}$ in the asynchronous generator (DG2), respectively. The block diagram of the voltage regulators is shown in Figure 7, where $\tau_1$ and $\tau_2$ are the parameters for the filters, and $V_{dc}^*$ and $V_{dc}$ reference the DC-link voltage and the measured DC-link voltage. Two filters are applied to smooth the input and output control signals by reducing the noise, with their parameters summarized in the Appendix A. $\varepsilon_{1,power}$ and $\varepsilon_{2,power}$ are the control signals from the secondary controller aiming to regulate the DGRS according to the optimal power set-points, which will be discussed in the next section. If $\varepsilon_{1,power}$ and $\varepsilon_{2,power}$ are equal to 0, this control loop would be the primary strategy for the bus voltage control in the diesel electric mode.

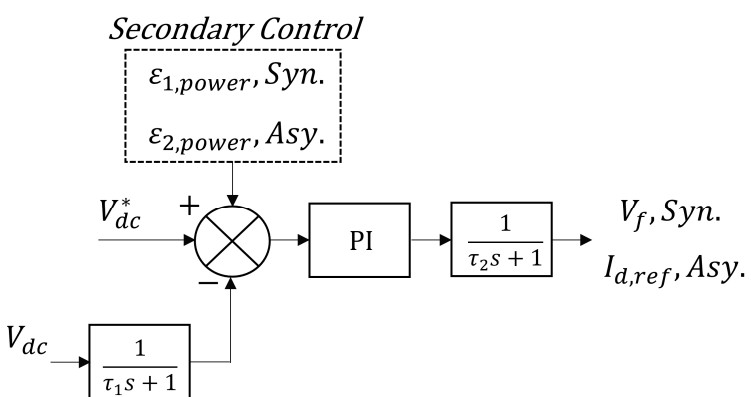

**Figure 7.** Voltage regulator of diesel generators.

(3) DC–DC Converter Control: A DC–DC converter controller is modelled and presented in Figure 8, as a voltage controller in hybrid mode. The actual DC-link voltage is regulated through its difference from the reference signal.

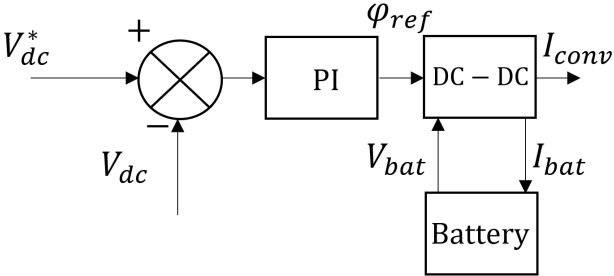

**Figure 8.** Control strategy of ESS.

For the model of the DC–DC converter presented in Figure 4, the duty cycle, phase angle shift between LVS and HVS voltages, φ, and switching frequency, $f_s$, are the three main control variables. By controlling any of these three or all of them together, power flow can be controlled in either direction [35]. Here, the duty cycle and switching frequency are fixed at constant values of 50% and 20 kHz, respectively. An adjustable leakage inductance $L_s$ is applied to regulate the phase shift φ, which is set as the unique control signal in this system. A PI controller is used to regulate the DC-link voltage according to the voltage error, and the controller will then generate the reference phase shift, $\varphi_{ref}$, for the converter to determine the mode of the battery (charge/discharge) according to the sign of the reference phase shift. The control system output DC current, $I_{conv.}$, can be generated to balance the target load current. A DC–DC converter has been applied to readjust the net current supplied by the power sources/energy storage devices into the DC bus, assuming a constant efficiency of 95% [32]. This is the primary strategy for bus voltage control in the hybrid mode.

*3.2. Energy and Power Management System*

As the DC shipboard hybrid power system has very high dynamics, the control system should also be fast acting. However, high load transients can make the system unstable. In order to reduce the stiffness of the system, PI controllers are used to adjust the power references from the Energy Management System (EMS). The secondary control strategy, PMS in this work, is achieved by applying an additional power control loop on top of the primary control shown in Figure 7.

$$e_{p,DGk} = P_{ref,DGk} - P_{DGk}, \; k \in \{1,2\} \tag{28}$$

$$\varepsilon_{k,power} = K_{pp,DGk} e_{p,DGk} + \frac{K_{pp,DGk}}{T_{ip,DGk}} \int e_{p,DGk} dt \tag{29}$$

where $e_{p,DGk}$ is the power error of DGRS, $P_{ref,DGk}$ and $P_{DGk}$ are the reference and measured DGRS power, respectively. $K_{pp,DGk}$ and $T_{ip,DGk}$ are the control parameters for the PI secondary controller.

As mentioned in the description of Figure 5, there are two modes involved in the proposed hybrid power system. In the hybrid mode, the ESS will play a role to regulate the DC-link voltage, and the DGRS sets will work at the optimal power set-points. Thus, the control signals are derived from Equation (29). In the diesel electric mode, the DGRS sets need to maintain the DC-link voltage, instead of ESS. In this condition, the secondary controller will be shut down. The on–off-switching state of the secondary controllers is presented in Table 1.

**Table 1.** On–off-switching of Secondary Controllers.

|  | Hybrid Mode | Diesel Electric Mode |
|---|---|---|
| Value of $\varepsilon_{k,power}$ | $\varepsilon_{k,power}$ obtained from Equation (22) | $\varepsilon_{k,power} = 0$ |
| Switching state of controller | State "ON" | State "OFF" |

The EMS acts as a tertiary level control system. In this work, the EMS model decides the operation of the generator depending on the priority selection of the power sources. Rule-based EMS strategies have been developed based on battery SOC% and real-time load power, which are given in Table 2. This EMS strategy is based on previous work presented in Zhou et al. [38]. The load power, the DGRS power reference, the minimum DGRS power, and the optimal DGRS power are denoted by $P_L$, $P_{DG, ref}$, $P_{DG,min}$, and $P_{DG,opt}$, respectively. Consequently, whether the battery is charging (*Ch.*) or discharging (*Dis.*) can be determined by the reference power settings of the DGRS.

**Table 2.** On–off-switching of Secondary Controllers.

| | $\mathbf{SOC\% \leq SOC\%_{min}}$ | $\mathbf{SOC\%_{min} < SOC\% < SOC\%_{max}}$ | $\mathbf{SOC\% \geq SOC\%_{max}}$ |
|---|---|---|---|
| $0 < P_L \leq P_{DG,opt}$ | $P_{DG, ref} = P_{DG,opt}$ <br> Battery Ch. | $P_{DG, ref} = \begin{cases} P_{DG,min}, & Dis. \\ P_{DG, opt}, & Ch. \end{cases}$ | $P_{DG, ref} = P_{DG,min}$ <br> Battery Dis. |
| $P_{DG,opt} < P_L \leq 2P_{DG,opt}$ | $P_{DG, ref} = 2P_{DG,opt}$ <br> Battery Ch. | $P_{DG, ref} = \begin{cases} P_{DG,opt}, & Dis. \\ 2P_{DG, opt}, & Ch. \end{cases}$ | $P_{DG, ref} = P_{DG,opt}$ <br> Battery Dis. |

### 3.3. Efficiency Optimization Algorithm

In conventional three-level control strategies, the optimal fuel consumption would be achieved when the engines are operated at their minimum SFC point. However, due to power losses in the energy storage, the optimal operation is not achieved by keeping the engine operating point at the minimum SFC during some specific loading conditions. In order to calculate the shifted optimal power set point of each of the power sources, an efficiency optimisation algorithm is developed to take the losses in the power components into consideration.

The first step of the optimisation process is to build a relationship between the actual DC source power from the diesel generator rectifier systems and hourly fuel consumption. Given k, the number of working engines, the relationship can be expressed as follows:

When k = 1,

$$C(P_{s1,k=1}) = c_0 D_{s1} + b_0 D_{s1} P_{s1,k=1} + a_0 D_{s1} P_{s1,k=1}^2 \tag{30}$$

When k = 2,

$$\begin{aligned} C(P_{s1,k=2}, \ P_{s2,k=2}) &= c_1 D_{s2} + b_1 D_{s2} P_{s2,k=2} + a_1 D_{s2} P_{s2,k=2}^2 + c_0(1 - D_{s2}) \\ &\quad + b_0(1 - D_{s2}) P_{s1,k=2} + a_0 (1 - D_{s2}) P_{s1,k=2}^2 \end{aligned} \tag{31}$$

where $P_{s1}$ and $P_{s2}$ are the DC-source powers for engine number 1 and engine number 2, respectively. $D_{s1}$ and $D_{s2}$ are the duty cycles for the one and two active engine condition, respectively. $c_0$, $b_0$ and $a_0$ are coefficients for the synchronous DGRS hourly fuel consumption. $c_1$, $b_1$, and $a_1$ are coefficients for hourly fuel consumption when two DGRS sets work together.

To balance the load power $P_L$ and the loss in the energy storage system during the charging and discharging processes, relationships can be derived between duty cycle, load power, and DGRS power, thus the duty cycle can be expressed as follows:

$$D_{sk} = \begin{cases} \dfrac{(2 - \eta_{converter})P_L}{P_{s1,k=1}\eta_{converter} + 2(1 - \eta_{converter})P_L}, & k = 1 \\[4mm] \dfrac{(2 - \eta_{converter})P_L - (2 - \eta_{converter})P_{s1,k=2}}{P_{s2,k=2}\eta_{converter} + 2(1 - \eta_{converter})P_L - (2 - \eta_{converter})P_{s1,k=2}}, & k = 2 \end{cases} \tag{32}$$

Assuming that $\eta_{converter}$ is a constant, given target $P_L$, the fuel consumption represented by Equations (30) and (31) has a minimum value that satisfies the following:

$$\begin{cases} \dfrac{\partial C(P_{s1,k=1})}{\partial P_{s1,k=1}} = 0, k = 1 \\ \dfrac{\partial C(P_{s1,k=2}, \ P_{s2,k=2})}{\partial P_{s1,k=2}} = 0, \dfrac{\partial C(P_{s1,k=2}, \ P_{s2,k=2})}{\partial P_{s2,k=2}} = 0, k = 2 \end{cases} \tag{33}$$

Based on the real-time load power $P_L(t)$, the optimal modes for the hybrid propulsion system, including charging/discharging and continuous modes, the corresponding working number of DGRS and reference power set-points for each engine can be obtained as control signals for the EMS through the algorithm shown in Figure 9.

Four operating modes according to the DGRS number k and the state of the ESS are discussed below:

(1) Ch/dis mode (k = 1): This mode is applied to relatively light loading conditions. The ESS will start firstly to compensate the load power. When battery SOC% goes below $SOC_{min}$, the DG1 will be powered up to balance the load and charge the battery. The DG2 will not be involved at this stage.

(2) Continuous mode (k = 1): As $P_L(t)$ increases, the duty cycle $D_{s1}$ for the Syn. DGRS will rise; thus, the power from the Syn. DGRS available to charge the battery decreases. When the duty cycle reaches 1, the battery SOC% will be stable as no power flows into the battery. At this stage, the Syn. DGRS will work alone to balance the load power.

(3) Ch/dis mode (k = 2): In this loading condition, the Asy. DGRS will be powered up to cooperate with the Syn. DGRS. The Syn. DGRS will always be kept on at this stage to match the high loading power. The Asy. DGRS will be involved only if the FC (fuel consumption) per hour of the Continuous mode (k = 1) $C_{min}(P_{s1,k=1}) >$ Ch/dis mode (k = 2) $C_{min}(P_{s1,k=2}, P_{s2,k=2})$, which means that the FC of one DGRS in operation is higher than that of two DGRS. The ESS will be charged/discharged according to the on/off state of the Asy. DGRS.

(4) Continuous mode (k = 2): In the condition of Ch/dis mode (k = 2), with the increase in load power, $P_L(t)$, the system will be switched to the Continuous mode (k = 2) when the duty cycle $D_{s2}$ gradually approaches 1. In this case, both the DG1 and DG2 are powered up and share the load power evenly. Simultaneously, the battery SOC% will be maintained at a steady level, as no power flows in/out of the ESS. This mode is applied to tackle the highest power-demand condition.

Moreover, the shifted power set-point for the power sources can be determined from the algorithm in different loading conditions. The operating modes of the hybrid propulsion system are determined by whether an extreme can be detected over the curve or surface built between DC power and hourly fuel consumption. As shown in Figure 9a,c, extremes, which are highlighted in yellow and green, can be detected and calculated by conducting partial differential analysis of $C_{min}(P_{s1,k=1})$ and $C_{min}(P_{s1,k=2}, P_{s2,k=2})$, respectively. When k = 1, the resulting point "optimal $P_{s1,k=1}$" in Figure 9a is used as the reference power set-point to regulate the power of the Syn. DGRS. When k = 2, the Syn. DGRS and Asy. DGRS will share the reference DC power "Optimal $P_{s2,k=2}$" evenly in the battery charging mode, as the parameters of the two DGRS systems are similar. In the battery discharging mode, the Asy. DGRS will be powered down whilst the Syn. DGRS works at "optimal $P_{s1,k=2}$". Continuous modes are applied in conditions where no extreme can be obtained. As presented in Figure 9b,d, the highlighted minima, rather than extremes, indicate that the duty cycle for the DGRS sets in both k = 1 and k = 2 conditions are 1. In this case, the Syn. DGRS will be controlled to balance load power alone in the condition of k = 1 (Figure 9b). When k = 2, the two DG sets will share the load power evenly (Figure 9d).

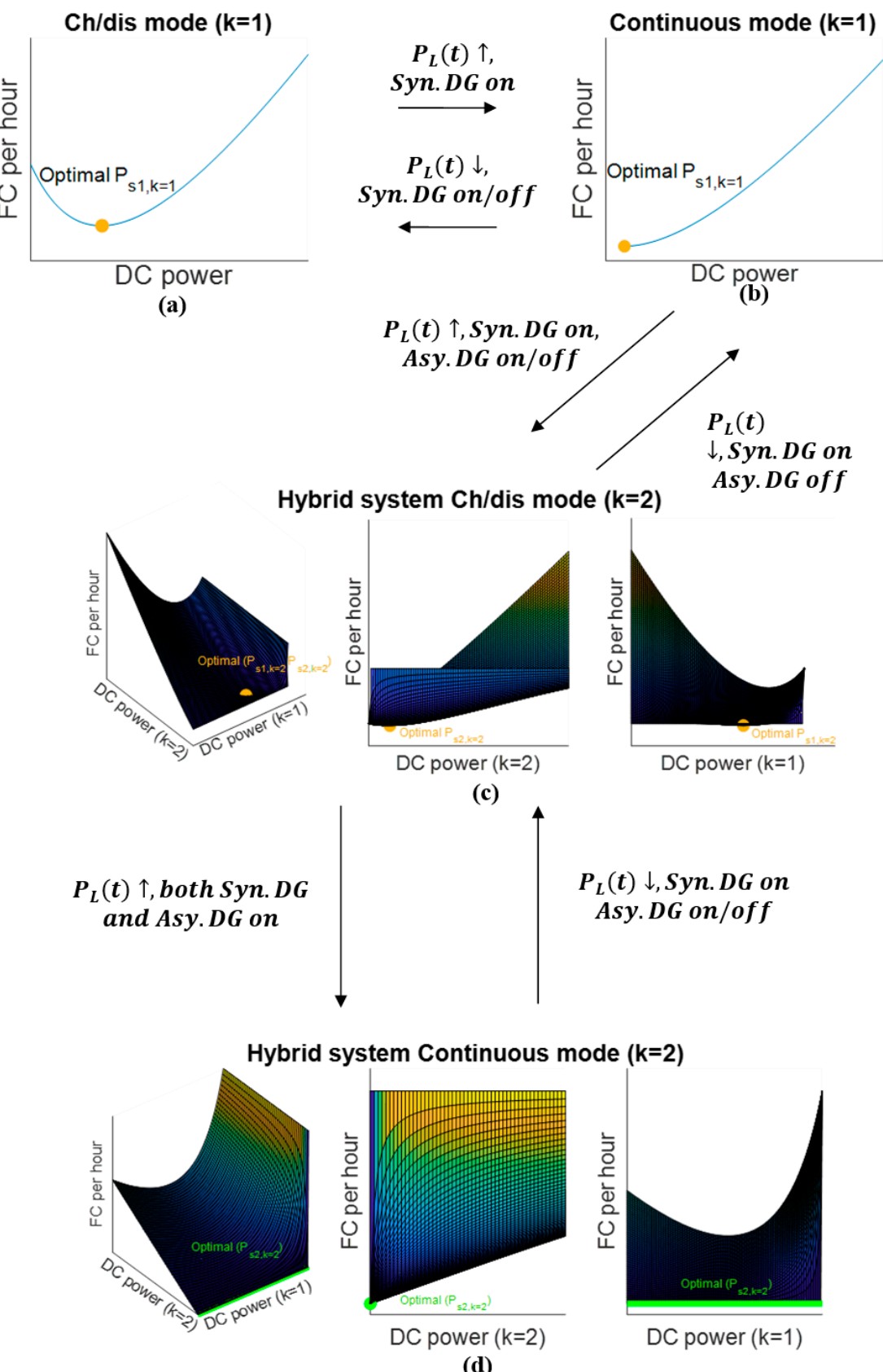

**Figure 9.** Algorithm of optimal mode determination for three-level control: (**a**) Ch/dis mode (k = 1), (**b**) Continuous mode (k = 1), (**c**) Ch/dis mode (k = 2), (**d**) Continuous mode (k = 2).

## 4. Simulation and Experimental Results in Optimisation Control

The complete system model and control strategies developed in the previous sections allows the simulation and experiments to be conducted accordingly. In this section, the simulation and experimental results of the DC hybrid power system with the proposed control strategies will be demonstrated and analysed. For the primary control, because of platform settings, the control strategy applied is an on–off control, which is different from the PID control strategy proposed in the modelling and simulation stage. However, the secondary and tertiary control with a proposed efficiency algorithm has been adopted in the PMS system. Thus, the platform is partially validated in the present work.

### 4.1. Simulation and Experimental Setups

(1) Load Profile: To set up a lab-scale experiment for simulating the ice loading condition (open water → heavy ice load → light ice load condition), the designed loading power needed to be varied widely. In this scenario, the load power for the experiments is designed in a range of 3.5 kW to 85 kW. Figure 10 presents the load profile for the simulation and the resulting load power in the experiment. The adjustable resistance is connected to the load side to regulate load power accordingly. At the start, the required load power is at a low level to simulate the ship operating in an open water part of the voyage. After 200 s, the required load power increases rapidly to 85 kW after a brief period of oscillation, to examine the power system response to a sudden change to high loading condition when, for example, the ship encounters thick ice. Finally, the required load power drops to 43 kW, corresponding to the ship encountering a thin ice loading condition.

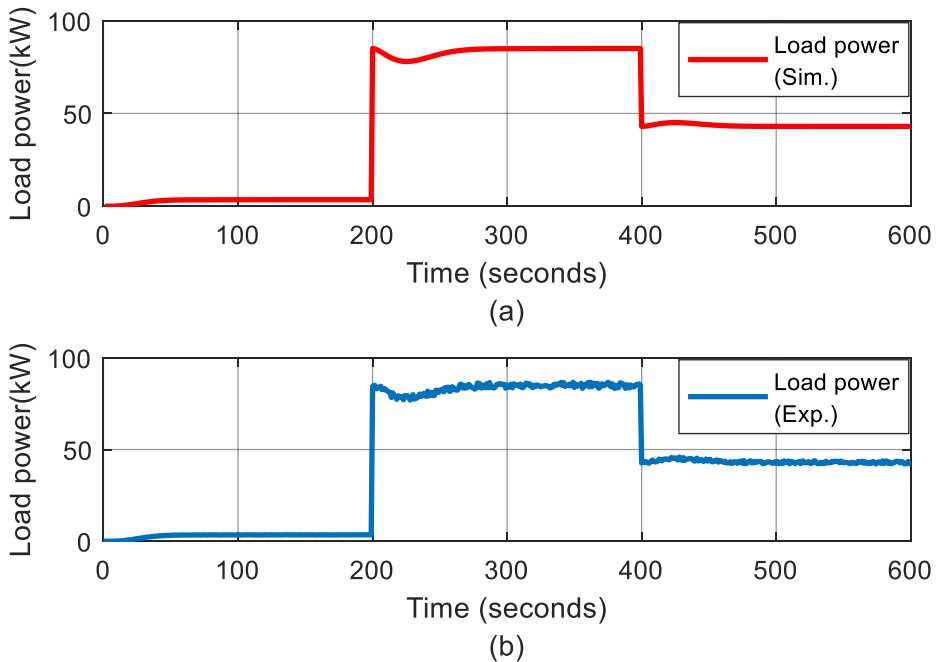

**Figure 10.** Load profile in scaled-down scenario. (**a**) Simulation load profile (**b**) Experiment load profile.

(2) Parameters of System Components: The proposed control method simulation is compared with a set of experimental results collected from the hybrid power lab in the Shanghai Marine Diesel Engine Research Institute (SMDERI). The configuration of the testbed is shown in Figure 11a, with a block diagram of the system with control stations given in Figure 11b. There are two sets of diesel engines and generators with rectifiers, and batteries, which are connected to a DC-bus through DC–DC converters. An adjustable resistance is applied to emulate the load profile shown in Figure 10. Detailed specifications of the equipment can be found in Table 3.

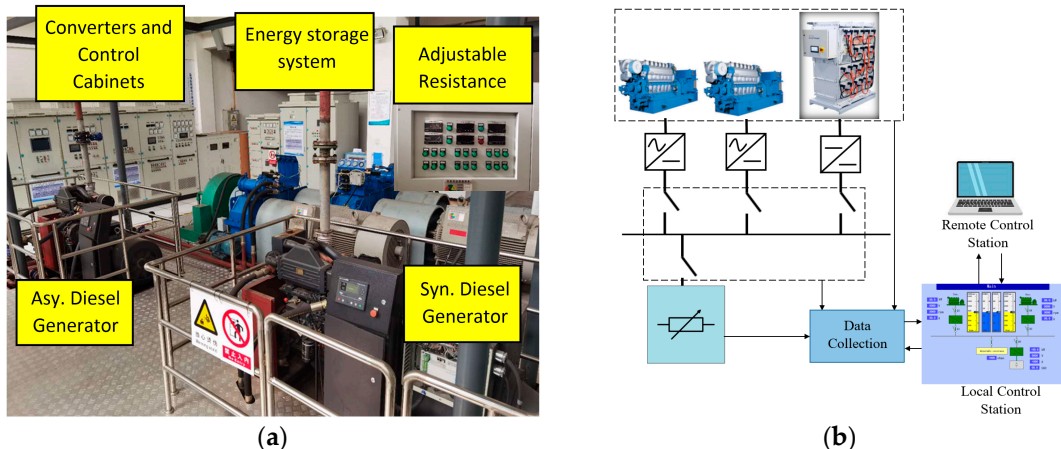

(**a**)           (**b**)

**Figure 11.** Laboratory facility in Hybrid Power Lab (**a**) Equipment Installation (**b**) Overview of the experimental setup, data collection, and control stations.

**Table 3.** Rated Values of Component Parameters.

| Components | Specifications |
|---|---|
| Syn.Generator | 4 poles, 800 rpm − 1600 rpm, 100 kW, 690 V |
| Asy.Generator | 4 poles, 800 rpm − 1600 rpm, 100 kW, 690 V |
| Energy Storage System | Lithium-ion, 384 V, 38 kW, $\eta_{ESS} = 95\%$ |
| DC-link Voltage | 1000 V |

In order to implement the efficiency optimisation algorithm, the power range diagrams with SFC and DC output power for the Syn. and Asy. DGRS had to be obtained from the manufacturer; these are presented in Figure 12a,b, respectively. The optimal operating points and the corresponding SFC for the DGRS sets in different speed and power ranges can be obtained from these diagrams. Based on the power range diagrams, the optimal power set-points of the DGRS sets can be derived from the proposed algorithm. In view of the conventional hierarchical control strategy, it can be concluded that the optimal power setpoint for both Syn. DGRS and Asy. DGRS is around 60 kW at a speed of 1500 rpm, without the efficiency of the ESS taken into consideration. The data acquisition unit was limited to storing 1000 points; therefore, the sample time for the data collection was set as 1 s for the 600-s profile.

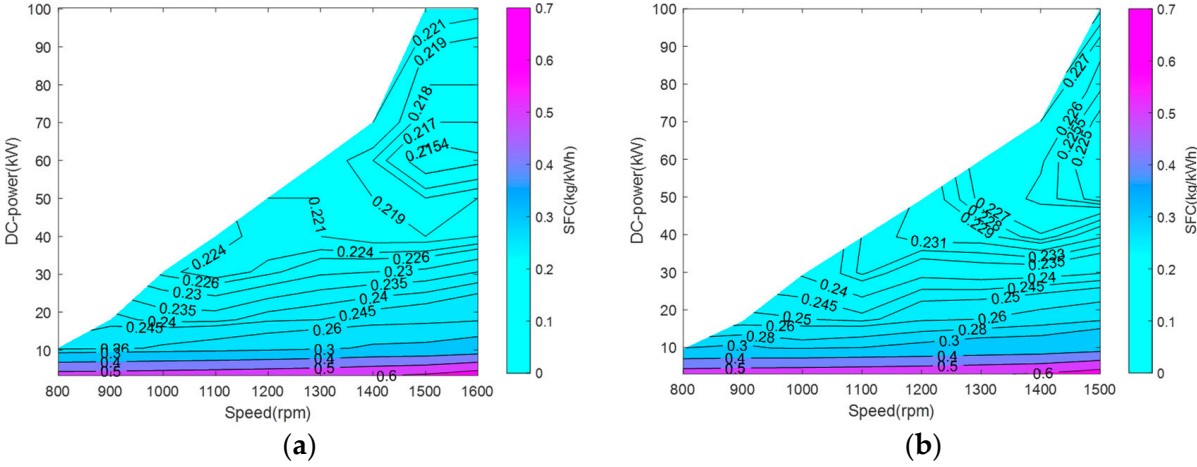

(**a**)           (**b**)

**Figure 12.** Power range diagram of (**a**) Syn. DGRS (**b**) Asy. DGRS.

### 4.2. Simulation and Experimental Results

The results from the simulation and experiments for the DC hybrid power system with the proposed control strategies discussed in Section 3 are compared and analysed in terms of DC-link voltage, DC current from Syn. DGRS, Asy. DGRS, and ESS, battery SOC%, and online DGRS number when given the same load profile. The overall comparison demonstrated that there are differences between the simulation and the experimental results mainly due to the different settings of the low-level control system of the converters in the laboratory, which cannot be fully accessed to duplicate those in the mathematical model. Moreover, components like filter inductors and capacitors are simplified in the simulation model, which adds to the difference in setups.

Figures 13 and 14 show the simulation and experimental results of the DC hybrid power system with the proposed control strategies. These figures demonstrate that the simulation results, based on the load condition in Figure 10, are valid as they match the experimental results closely, even given the difference in the primary controllers, essentially because the aim of the controllers is the same, i.e., to regulate the subsystem outputs according to the reference signals. The experimental results for the DC-link voltage are presented in Figure 13, and it is clear that a relatively high spike (9.5%) is generated due to the change in DG operation modes (hybrid propulsion mode ↔ diesel electric propulsion mode).

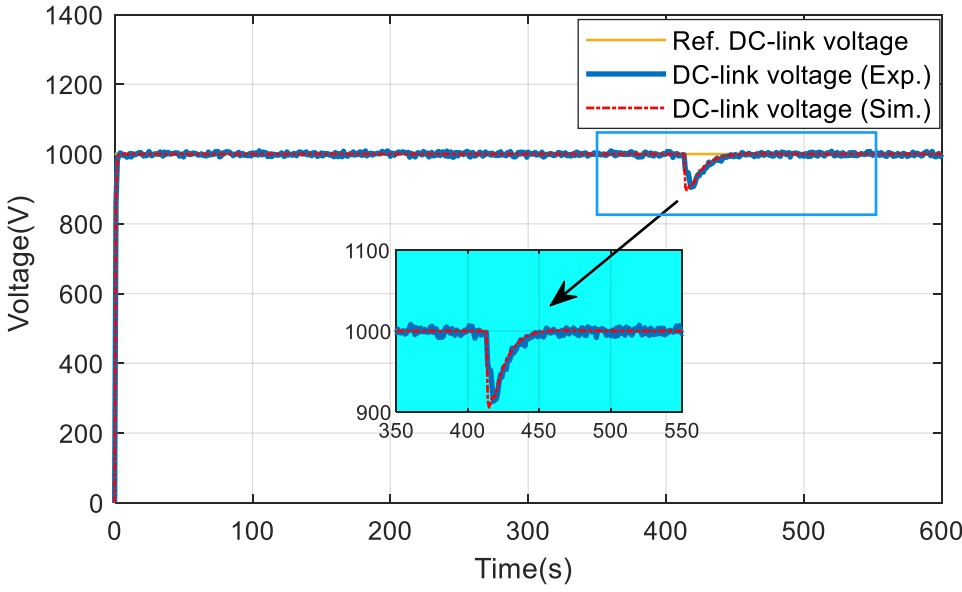

**Figure 13.** Comparison between the experiment and the simulation in DC-link voltage.

Over the first 200 s, as shown in Figure 14a–c,e, only the ESS will provide power to balance the relatively low load power. Due to the increase in load power (from 3.5 kW to 85 kW) at 200 s, the DG1 is powered up to supply optimal $P_{s2,k=1}$ until the battery SOC%, as shown in Figure 14d, hits the lower boundary at 212 s. In order to demonstrate the process of battery charging and discharging, battery SOC% variations are limited to be between 64.1% and 65.3%, as the capacity of battery in the lab is relatively high for the 600 s time period. At 212 s, the DG2 is powered up with charging of the battery. The DG1 and DG2 operate together and share the optimal power setpoint $P_{s2,k=2}$ evenly to balance the high-level load. According to Figure 14e, both the DGs are disconnected from the grid due to the load change from 85 kW to 43 kW at 400 s. During the first 13 s period after the load change, the ESS is ordered to work solely until its SOC% drops to the lower boundary. Then the DG1 is powered up to balance the load independently until the end of the scenario, in the continuous mode (k = 1).

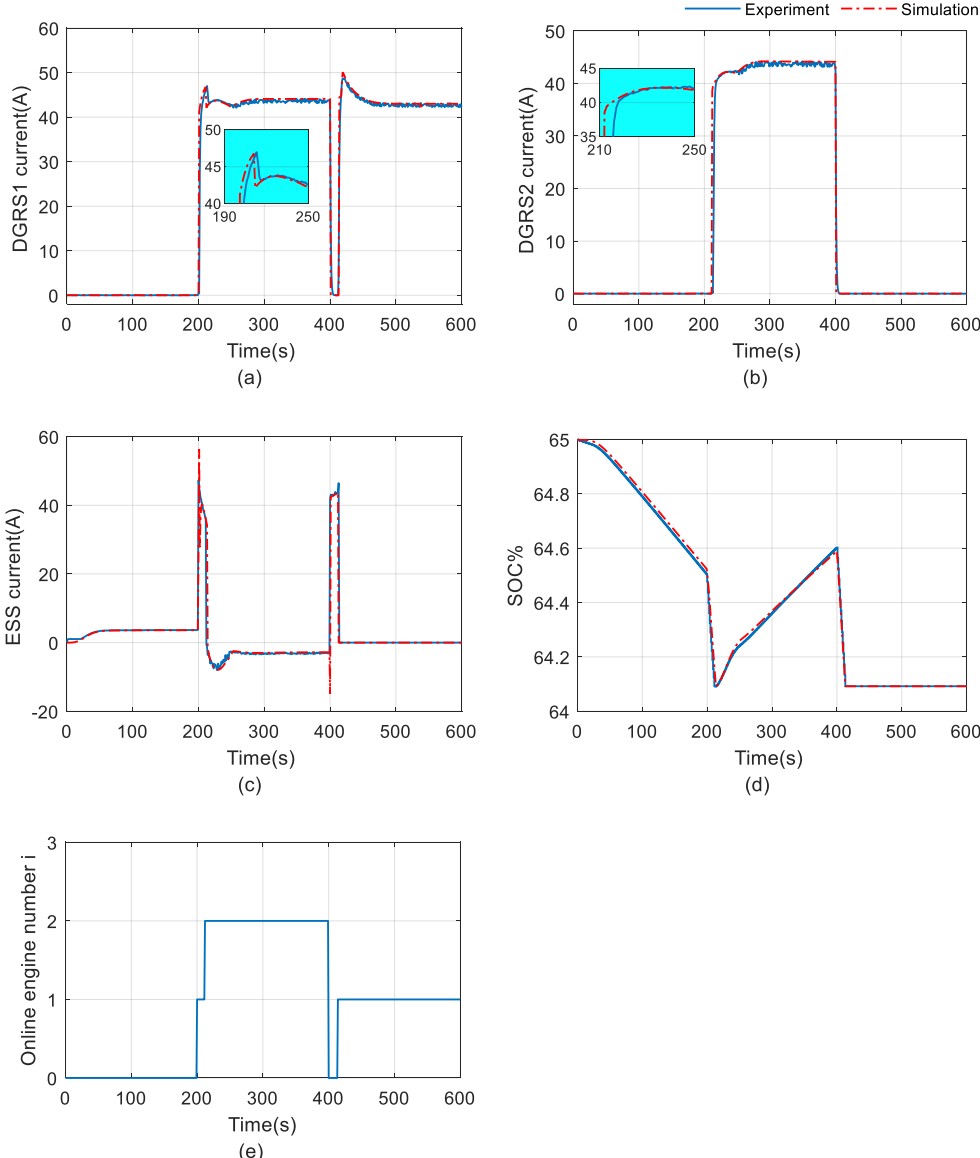

**Figure 14.** Comparison between the experiment and the simulation in (**a**) Syn. DGRS current, (**b**) Asy. DGRS current, (**c**) ESS current, (**d**) Battery SOC%, (**e**) Online DGRS number.

It is worth noting that the experimental results showed some delay when compared with the simulation results in terms of the DGRS output of the proposed efficiency-optimized hybrid power system demonstrated, this can be seen in the zoomed-in views of Figure 14a,b. The observed discrepancies in the experimental platform can likely be attributed to various factors, including the time delay caused by the inertia of real machines and the cooperation between the ESS and DGRS. Additionally, differences in the primary control strategy employed could also contribute to variations between the experimental and simulation results. This phenomenon finds support in the similar work conducted by Chua et al. [37]. Their findings demonstrate certain delays in the experimental signals compared with the simulation results, particularly regarding the output power percentage signals of the DGRS. This observation aligns with the outcomes reported in the present study. Despite these disparities, the control system effectively guided the variables towards their respective reference values after the time delay. Moreover, the battery capacity in Chua et al.'s research is quite high when compared with the scenario load power (Maximal load power around 6.5 kW) and battery power (Maximal battery power around 1.7 kW), so the battery SOC% in their research was relatively stable during the change in battery

modes. Apart from the time delay problem in the experimental results and the ripples in the simulation results, it can be concluded that the feasibility of the proposed control strategies in this work have been clearly demonstrated by both simulation and experimental results.

## 5. Comparison of Various System Configurations

To further investigate the proposed control strategy, lab-scale experiments were also designed to evaluate the performance of a DC diesel electric system with the engine operating at a fixed-speed and at a variable-speed, and a hybrid system with a conventional three-level control strategy. The system performance was again tested in a simulated scenario with variation in ice loading. The setups of the different system configurations and control theories are detailed below:

DC diesel electric system with fixed- and variable-speed operations: The optimal selection of an engine operating speed for fixed-speed and variable-speed DC diesel electric systems can be derived based on the power range diagrams, as shown in Figure 15. For the DC diesel electric system, only Syn. DGRS will be operated to balance the load power. With the primary control from the engine governor shown in Figure 6 and the voltage regulator given in Figure 7, the output of the Syn. DGRS can be regulated to balance the changing load power.

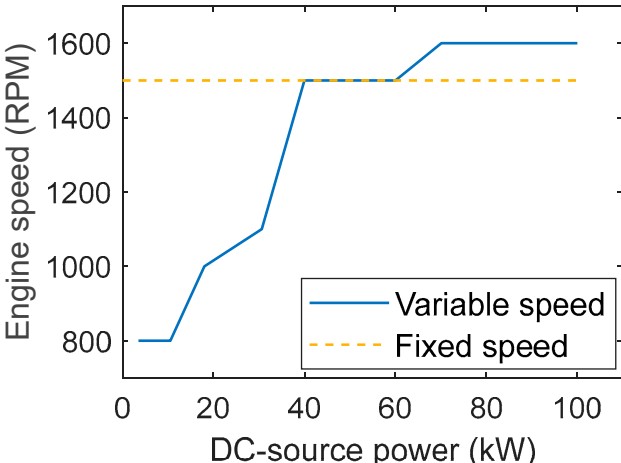

**Figure 15.** Optimal speed selection.

Hybrid system with conventional three-level control strategy: For the conventional three-level control strategy, the DGRS sets are seeking to work at the minimal SFC points. Thus, the optimal power setpoint for both Syn. DGRS and Asy. DGRS is around 60 kW at a speed of 1500 RPM. According to the optimal power setpoints obtained, the rule-based EMS Strategy in Table 2 can be implemented with $P_{DG,opt}$ set as 60 kW. Then, the operating modes can be determined according to load power $P_L$ and battery SOC%.

Firstly, the fixed and variable speed DC diesel electric systems are evaluated using the scenario set out in Figure 10. In this case, the power system will be connected to the DC grid to balance the load power at the start. As shown in Figure 16, the system requires about 75 s to settle down to steady state (1000 V) due to load power variation, which indicates that the DC diesel electric systems are sensitive to load change. In the diesel electric system of this experiment, only the Syn. DGRS is involved as its rated power is 100 kW, which is able to cover the load power during the scenario. Both the fixed- and variable-speed operations provide acceptable voltage error of around +/−5% which would be within the regulation limit for the DC bus according to Kim et al. [39].

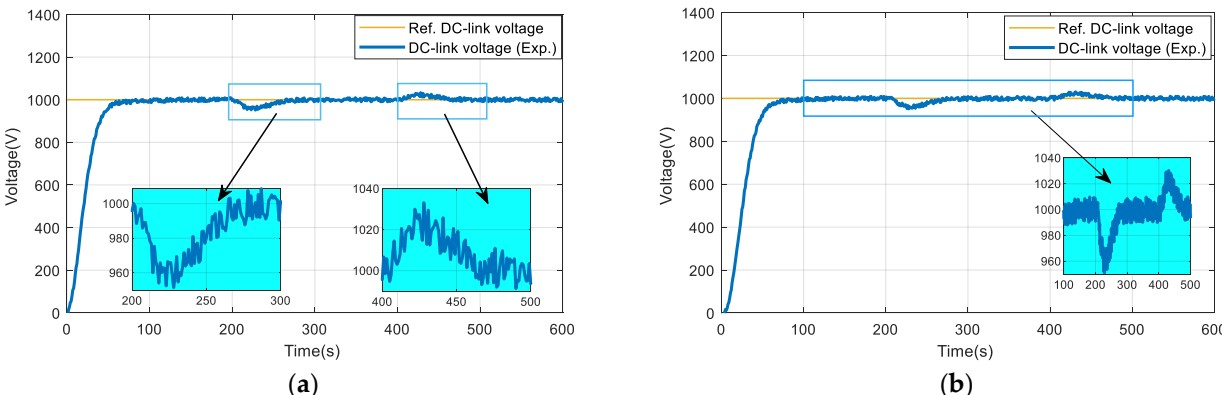

**Figure 16.** Experimental results of (**a**) Fixed-speed diesel electric system DC-link voltage, (**b**) Variable-speed diesel electric system DC-link voltage.

Figure 17 presents a comparison in terms of DGRS output current and engine speed for fixed versus variable speed operation. As can be seen in Figure 17a,b, both fixed-and variable-speed operations can track the load current in a steady state. The difference between the operations is the engine speed where, in variable speed operation, the engine speed can be regulated to achieve minimal fuel consumption at each power range. As shown in Figure 17c,d, in the three different power conditions, the engine is regulated to work at a speed of 800 rpm at 3.5 kW, 1600 rpm at 85 kW, and 1500 rpm at 43 kW, which is in accordance with the operational speed selection presented in Figure 15.

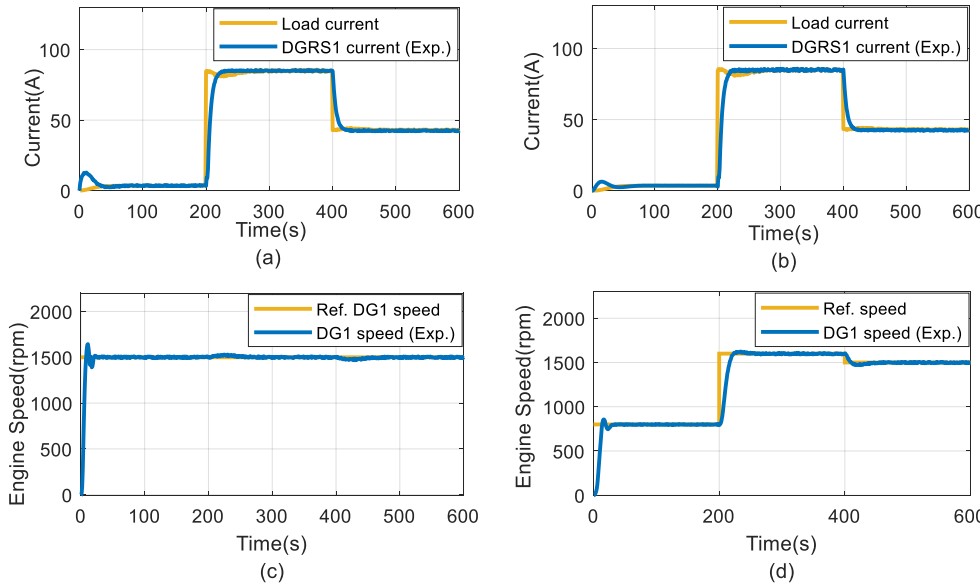

**Figure 17.** Experimental results of (**a**) Fixed-speed diesel electric system DGRS current (**b**) Variable-speed diesel electric system DGRS current (**c**) Fixed-speed diesel electric system engine speed (**d**) Variable-speed diesel electric system engine speed.

The experimental and simulation results of the conventional three-level control strategy and the proposed efficiency-optimized hybrid power system are presented in Figures 18–20. In the conventional hierarchical control hybrid system, the ESS will be involved and will play a role in regulating the DC-link voltage. Thus, as shown in Figure 18, the settling time of the hybrid power system to a steady state is reduced greatly compared with the DC diesel electric system through the assistance of the ESS. Figure 18 indicates that the DC-link voltage of the DC hybrid system is more sensitive to the change in battery operating modes (charging/discharging), rather than the variation in load power, as in the

case of the DC diesel electric system. Figure 18a demonstrates the results of the DC-link voltage in the conventional EMS; the spikes in the simulation results (highlighted in red) are within 0.08% of the nominal value. Due to the large sample time (1 s), the minor changes caused by the load and battery mode change in DC-link voltage cannot be observed in the experimental results (shown in blue). This indicates that the DC-link voltage settles down to steady state within a 1 s period. As shown in Figure 18b, a voltage transient is induced with a magnitude of 9.5% at 413 s, where the DC-link voltage of the proposed efficiency-optimized system is regulated by the voltage regulator of the DGRS to achieve the optimal operation in the target loading condition. However, Classification Society regulations limit the allowable voltage tolerance to ±10% so this would be acceptable [39]. The DC-link voltage reached the steady state again after 42 s, which proves that the voltage regulator in the DGRS set is able to generate the required stabilizing efforts due to a DGRS mode change.

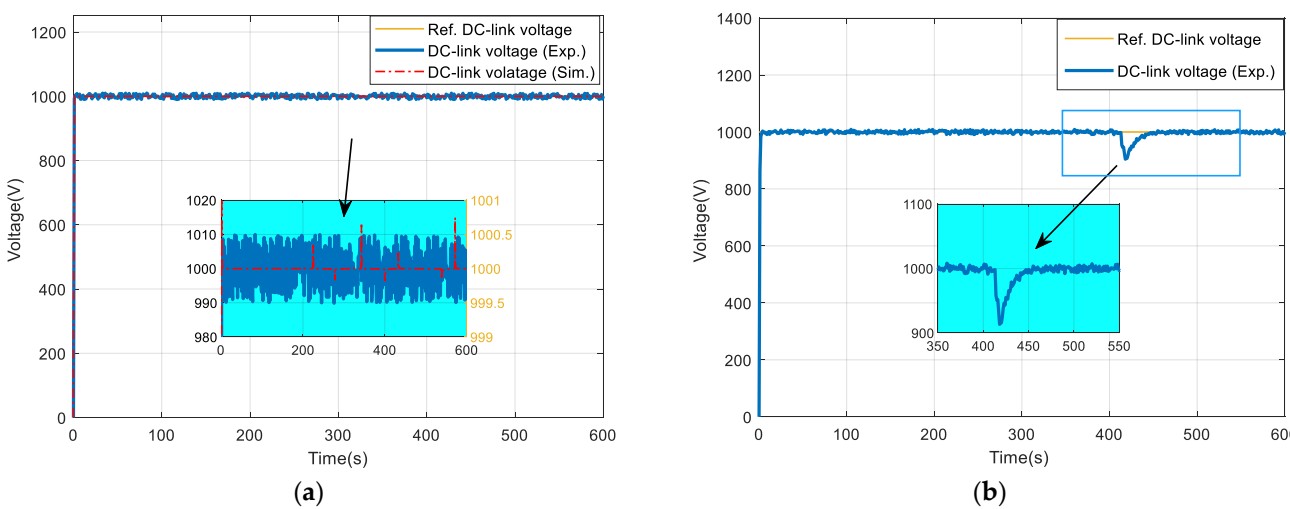

**Figure 18.** Experimental and simulation results of conventional three-level controlled hybrid power system: (**a**) DC-link voltage and the proposed optimally controlled hybrid system, and (**b**) DC-link voltage.

For the low load condition in the first 200 s, the strategic decisions of the conventional and efficiency-optimized control strategies are the same and only the ESS will provide power to balance the relatively low load power, as shown in Figure 19c,d,g,h. The difference can be observed after 200 s at which point, as shown in Figure 19c,e, the output current of the DGRS sets in the conventional three-level controlled system is fixed when connected to the grid and the whole system will operate according to the rule-based EMS strategy shown in Table 2. In contrast, the output of the DGRS sets in the proposed efficiency-optimized hybrid system, shown in Figure 19d,f is varied continuously according to the real-time load power. When the load power maintains 85 kW, the optimal power set-point for DGRS 1 is 44.08 kW, while the optimal power set-point for DGRS 2 reaches 44.16 kW, to achieve the optimal fuel consumption rate of 18.30 kg/h. In addition, the diesel electric mode is activated by the proposed control strategy after 400 s to supply the 43 kW load power. The battery SOC% in Figure 20b is maintained at a fixed value, while the SOC% under conventional control, shown in Figure 20a, is changing in a range between the SOC% boundaries. As shown in Figure 21, the duty cycle of the operating modes during the scenario has been presented. During the time period between 200 s and 400 s, the system is working in mode Ch/dis mode ($k = 2$), the DGRS 1 will always be kept on at this stage to match the high loading power with the duty cycle equals to 1, and the figure presents the duty cycle when two DGRS are working together. It can be found that the duty cycle varies in different loading conditions, which ranges from around 0.83 to 0.94. After 400 s, the

operating mode switches to Continuous mode ($k = 1$). In this case, only DGRS 1 is working with the duty cycle equal to 1.

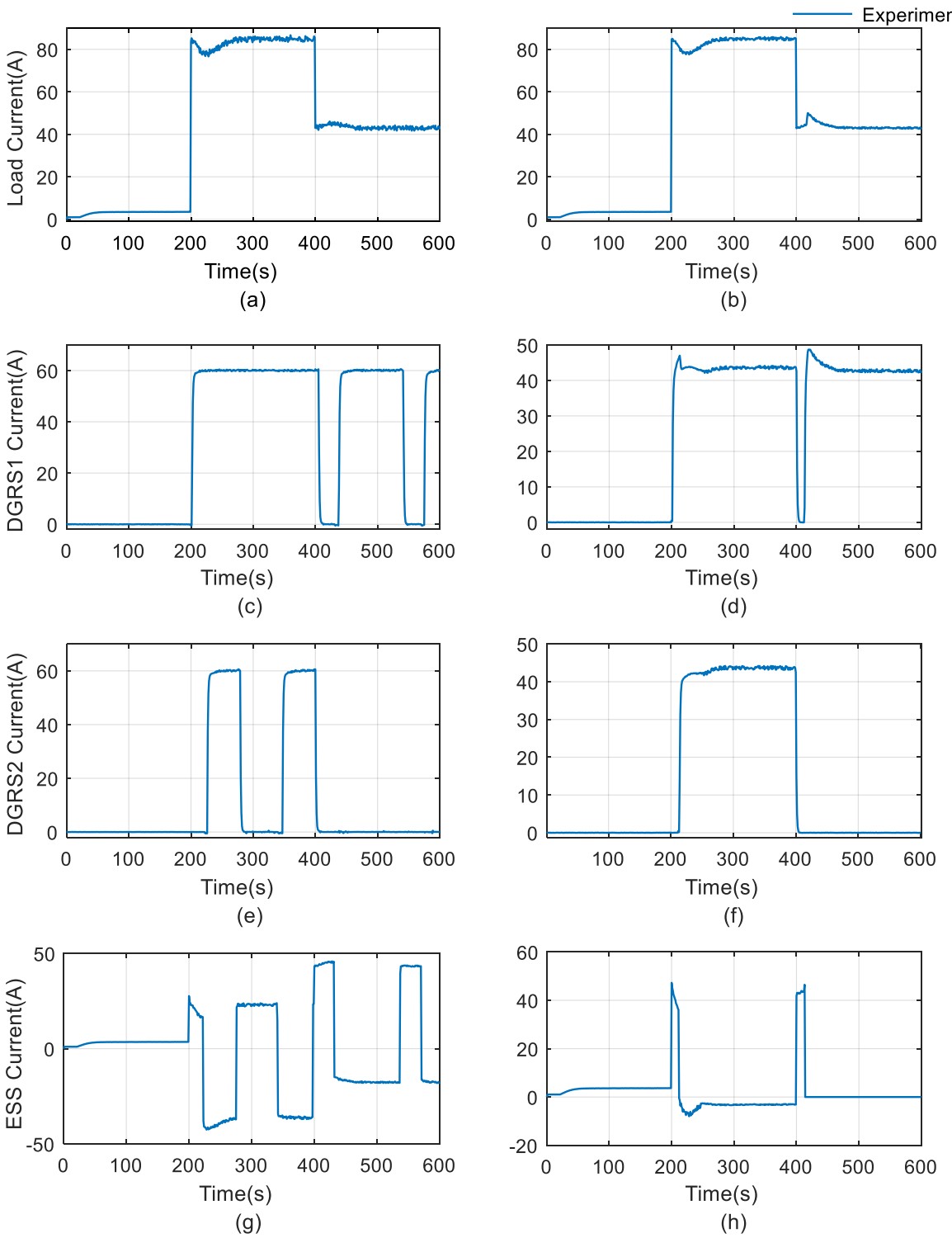

**Figure 19.** Experimental results of conventional three-level controlled hybrid power system: (**a**) Load current (**c**), Currents from DGRS1, (**e**) Currents from DGRS2, (**g**) Currents from ESS and the proposed optimally controlled hybrid power system, (**b**) Load current, (**d**) Currents from DGRS1, (**f**) Currents from DGRS2, and (**h**) Currents from ESS.

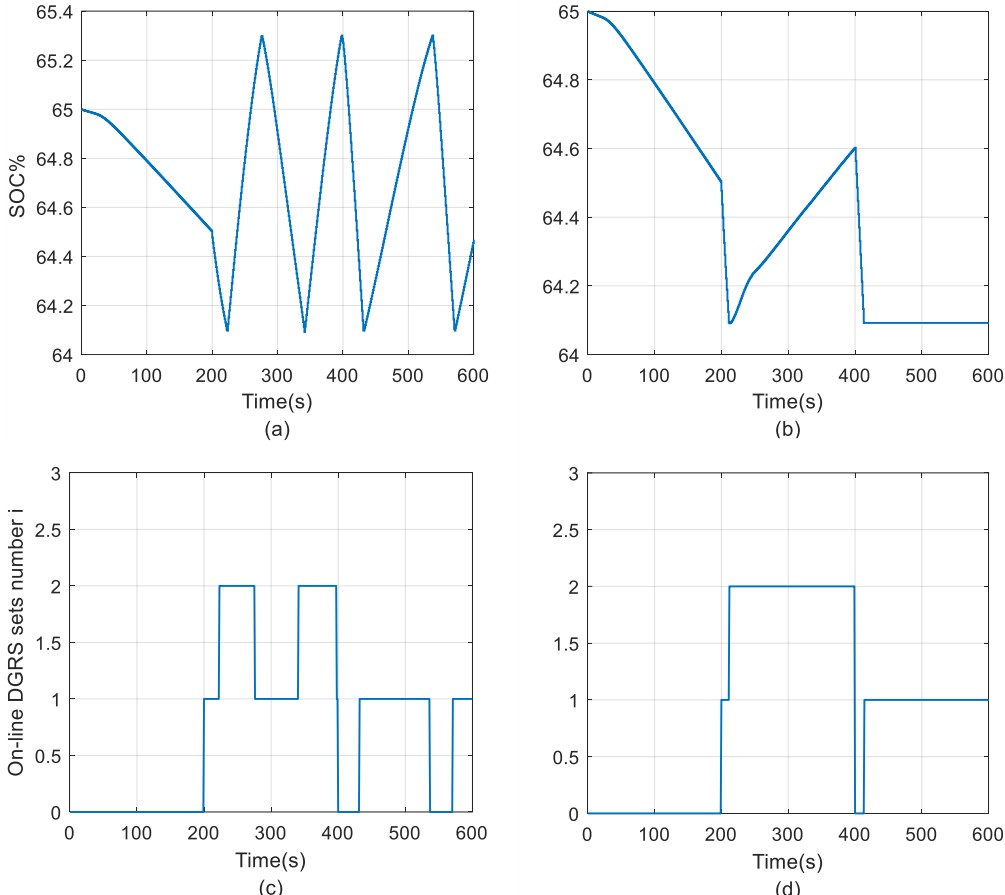

**Figure 20.** Experimental results of conventional three-level controlled hybrid power system: (**a**) Battery SOC%, (**c**) Online DGRS number and the proposed optimally controlled hybrid power system, (**b**) Battery SOC%, and (**d**) Online DGRS number.

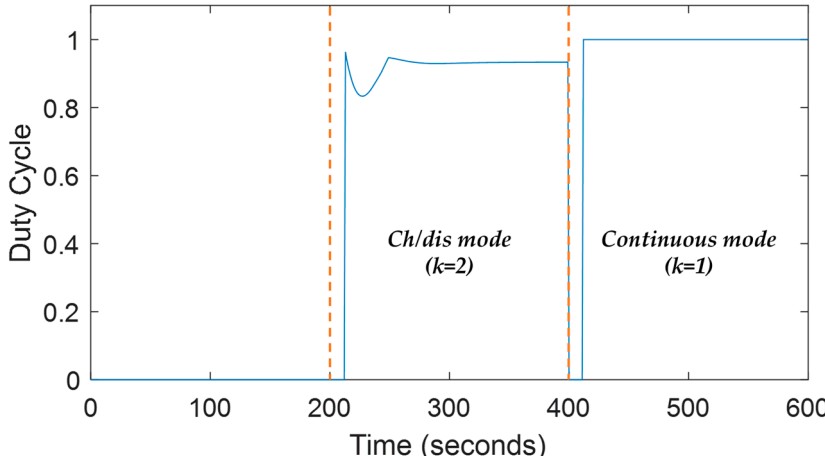

**Figure 21.** Duty cycle for DGRS of the proposed optimally controlled hybrid power system.

Due to the lack of equipment in the laboratory to measure fuel consumption, the resulting fuel consumption of the four configurations is calculated based on the output power of the DGRS sets, the power range diagrams shown in Figure 12, and the resulting battery SOC%. Figure 22 compares the fuel consumption of the systems with no net energy change in the battery SOC% during the scenario. The fuel oil consumption rate can be calculated through Figure 12 for the two DGRS by using the DC power in kW to multiply SFC in kg/kWh, with the battery SOC% and its equivalent fuel consumption rate

included in the charging periods for hybrid configuration. Then, the integral of the rate with respect to time (600 s) will give the fuel consumption for the two DGRS. For the diesel electric propulsion systems, compared with the fixed speed DC diesel electric system's fuel consumption (1.626 kg), the variable speed operation (1.610 kg) achieved a 0.98% fuel consumption reduction over the target journey. In terms of the hybrid propulsion system, despite the fact that an undesirable voltage transient was induced due to the change in DG operation modes, the proposed efficiency optimized hybrid propulsion system (1.539 kg) offers a 3.81% fuel saving over the conventionally controlled hybrid system (1.600 kg). Overall, the proposed efficiency-optimized system reduces the fuel consumption by 5.35% within the target journey when compared with the fixed-speed DC diesel electric system.

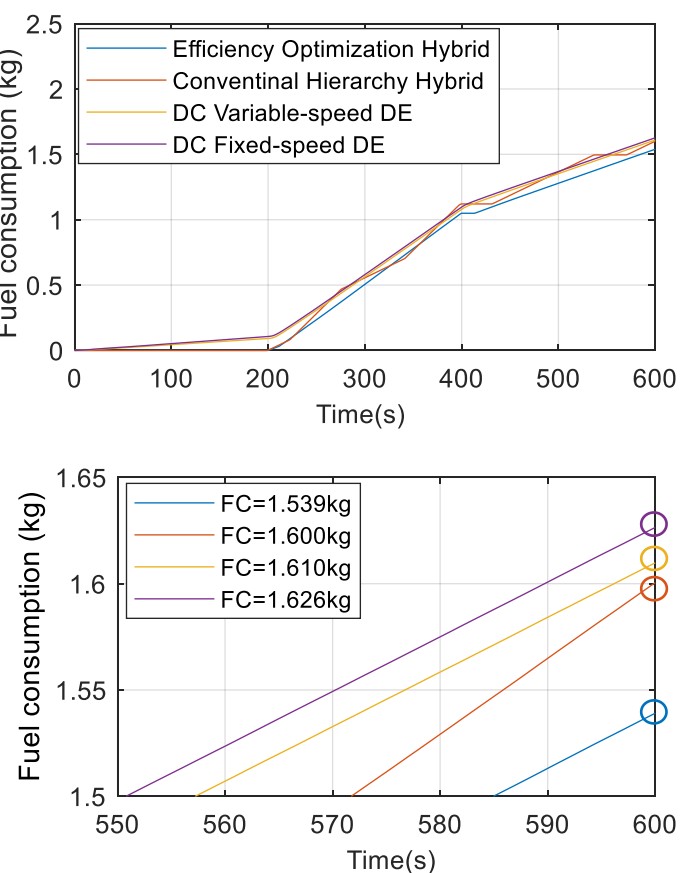

**Figure 22.** Experimental results of fuel consumption.

## 6. Conclusions

In this paper, system components of a DC shipboard power system have been mathematically modelled and a three-level control strategy has been developed based on the proposed hybrid DC power system. An efficiency optimisation algorithm has been proposed rather than a rule-based EMS to further consider the losses in the ESS. In addition, the modelled system has been tested with a scaled-down NSR shipping load profile in simulations and experiments. The system level testing has shown that the mathematical model is accurate enough to estimate the dynamics of a real system.

To further verify the superiority of the DC shipboard hybrid power system with the proposed control strategy, the small-scale experiments were designed to evaluate the performances of a DC diesel electric system with engine fixed-speed and variable-speed operations, and a hybrid system with rule-based and efficiency-optimized EMS. The algorithms for the different system configurations and control theories were established according to the parameters of the system components in the lab.

The test results demonstrated that for the hybrid propulsion system, the inclusion of energy storage resulted in a shorter settling time to a steady state compared with the DC diesel electric system. In addition, the disturbances in the DC-link voltage due to load change were significantly reduced in the hybrid propulsion system.

In terms of fuel consumption, the proposed efficiency optimized hybrid propulsion system achieved a fuel saving of 5.35% compared with the fixed-speed DC diesel electric system, and a 0.98% reduction achieved by the variable speed operation of the DC diesel electric system. This confirmed the significance and feasibility of the improvement in the configuration and control strategy for future NSR shipping.

For future work, the sample time for acquiring experimental data should be reduced to capture the variation in measured data due to load changes more accurately. Moreover, it is worthwhile noticing that the variation in DG modes applied in the proposed EMS would cause higher fluctuation in DC-link voltage when compared with the rule-based EMS; a comprehensive voltage stabilizing strategy is required to reduce the voltage transient variation during the change in operation modes. Moreover, due to the limitation of the lab settings, the proposed control strategy cannot be fully validated. Further works on an experimental platform that facilitates PID controllers will be conducted to match the theoretical analysis conditions as closely as possible.

**Author Contributions:** Conceptualization, Y.Z., K.P. and R.N.; methodology, Y.Z.; software, Y.Z.; validation, Y.Z.; formal analysis, Y.Z.; investigation, Y.Z.; resources, Y.Z., H.G. and Z.L.; data curation, Y.Z.; writing—original draft preparation, Y.Z.; writing—review and editing, Y.Z., K.P. and R.N.; visualization, Y.Z.; supervision, K.P. and R.N. All authors have read and agreed to the published version of the manuscript.

**Funding:** This research received no external funding.

**Institutional Review Board Statement:** Not applicable.

**Informed Consent Statement:** Not applicable.

**Data Availability Statement:** Not applicable.

**Conflicts of Interest:** The authors declare no conflict of interest.

## Nomenclature

The following abbreviations are used in this manuscript:

| | |
|---|---|
| $A$ | Exponential zone amplitude |
| $B$ | Inverse of the exponential zone time constant |
| $C$ | Fuel consumption per hour |
| $D$ | Damping coefficient of rotor |
| $D_{s1}, D_{s2}, D_{sk}$ | Duty cycles for the one and two active engine condition |
| $E_0$ | Battery constant voltage |
| $E_f$ | Field voltage vector |
| $I_{bat}, i_{bat}$ | Battery current |
| $I_{dc}$ | DC current to DC bus |
| $I_{conv.}, i_{dc-converter}$ | DC current from converter |
| $I_{d,ref}$ | Reference d-axis current |
| $I_{as}, I_s, I_{kf}, I_r$ | Current vector |
| $J$ | Inertia of generator |
| $K$ | Polarisation constant |
| $K_a$ | Actuator gain |
| $K_{pp,DGk}, T_{ip,DGk}$ | Control parameters for secondary controller |
| $L_{dc}$ | Inductance on the low voltage side of DC–DC converter |
| $L_S$ | Adjustable inductance in converter |
| $NO_2$ | Nitrogen Dioxide |

| | |
|---|---|
| $O_3$ | Ozone |
| $P$ | Number of poles |
| $P_{ref}$ | Reference power setpoint |
| $P_{DG}$ | Actual power for diesel generator rectifier system |
| $P_{DG,min}$ | Minimum diesel generator rectifier system power setpoint |
| $P_{DG,opt}$ | Optimal power setpoint of diesel generator rectifier system |
| $P_{DG,ref}, P_{ref,DG}$ | Reference power setpoint of diesel generator rectifier system |
| $P_{load}, P_L$ | Load power |
| $PMS$ | Power management system |
| $P_{out}$ | Output power |
| $P_{s1}, P_{s2}$ | DC-source powers for diesel generator rectifier system 1 and 2 |
| $Q$ | Battery capacity |
| $R, R_{bat}$ | Battery internal resistance |
| $R_{ar}, R_{as}, R_s, R_{kf}$ | Resistance matrices |
| $R_{load}$ | Resistance in the load side of DC–DC converter |
| $SOC\%$ | Battery state of charge |
| $SOC\%_{max}$ | Battery state of charge maximal threshold |
| $SOC\%_{min}$ | Battery state of charge minimal threshold |
| $T_1$ | Time constant of the actuator |
| $T_e$ | Electric torque |
| $T_m$ | Mechanical torque |
| $T_S$ | Sample time of converter |
| $U_a, U_s$ | Voltage vector |
| $V_{12}$ | Average voltage on the Low Voltage Side |
| $V_{34}$ | Average voltage on the High Voltage Side |
| $V_{Bat}$ | Battery voltage |
| $V_{dc}$ | DC-link voltage |
| $V_{dc}^*$ | Reference DC-link voltage |
| $V_f$ | Excitation field voltage |
| $V_s$ | Voltage on the load side of DC–DC converter |
| $X_{as}, X_s, X_{kf}, X_r$ | Leakage reactance matrices |
| $a_0, b_0, c_0$ | Coefficients for synchronous diesel generator rectifier system hourly fuel consumption |
| $a_1, b_1, c_1$ | Coefficients for hourly fuel consumption when two diesel generator rectifier sets work together |
| $e_{p,DGk}$ | Power error of diesel generator rectifier system |
| $f_s$ | Switching frequency of converter |
| $i_{dc}$ | DC current from rectifier |
| $i_{dqs}^r$ | dq-axis components of stator winding phase current |
| $i_{load}$ | Load current |
| $i^*$ | Filtered current |
| $It$ | Actual battery charge |
| $K$ | Number of diesel generator rectifier sets |
| $t_d$ | Time-delay constant |
| $u_c$ | Control signal from the engine controller |
| $u_i$ | Permutation matrix group |
| $v_{dc}$ | DC voltage of diesel generator rectifier system |
| $v_{dqs}^r$ | dq-axis components of stator winding phase voltage |
| $x_M$ | Integrated reactance |
| $x_{MD}, x_{MQ}$ | Integrated dq reactance |
| $\varepsilon$ | Speed error |
| $\varepsilon_{1,power}, \varepsilon_{2,power}, \varepsilon_{k,power}$ | Control signal from the secondary controller |
| $\eta_{converter}, \eta_{ESS}$ | DC–DC converter efficiency |
| $\tau_1, \tau_2$ | Parameters for the filters in voltage regulator |
| $\varphi$ | Phase shift regulated by the adjustable inductance |
| $\varphi_{ref}$ | Reference phase shift |
| $\psi_{as}, \psi_s, \psi_{kf}, \psi_r, \psi_{m1}$ | Magnetic flux vector |
| $\psi_m$ | Magnetizing flux vector |

| $\omega$ | Actual speed |
|---|---|
| $\omega_b$ | Base speed |
| $\omega_{ref},\omega^*$ | Speed reference |
| $\omega_{rotor}$ | Rotor speed |
| $\omega_e,\omega_{sr}$ | Speed matrix |
| $\omega_s$ | Slip matrices |
| $\Delta P_{ref,DG}$ | Power shift on reference power of diesel generator rectifier system by applying efficiency optimisation system |

## Appendix A

**Table A1.** DGRS parameters.

| Synchronous generator: | |
|---|---|
| Stator resistance $r_s$ = 0.382 Ω | Stator leakage reactance $x_s$ = 0.4222 Ω |
| Base electrical angular speed $\omega_b$ = 3600 rpm | |
| Field winding resistance $r_f$ = 0.112 Ω | Field winding reactance $x_f$ = 0.5768 Ω |
| Damping dq-axis winding resistance $r_{kd}$ = 14 Ω, $r_{kq}$ = 5.07 Ω | |
| Damping dq-axis winding reactance $x_{kd}$ = 3.7209 Ω, $x_{kq}$ = 9.3871 Ω | |

| Asynchronous generator: | |
|---|---|
| Stator resistance $r_{as}$ = 0.382 Ω | Stator leakage reactance $x_{ls}$ = 0.4222 Ω |
| Base electrical angular speed $\omega_b$ = 3600 rpm | |
| Rotor winding resistance, $r_{ar}$ = 0.11 Ω | Rotor reactance $x_{lr}$ = 0.57 Ω |
| Integrated reactance $x_M$ = 0.09 Ω | |

| Filter parameters: |
|---|
| $\tau_1 = 0.001,\ \tau_2 = 0.02$ |

| Mechanical system: | |
|---|---|
| Inertia moment J = 0.03 kg·m$^2$ | Damping coefficient D = 0.85 kg·m$^2$/s |

| Diesel engine parameters: | |
|---|---|
| Actuator gain $K_a$ = 1.5 × 10$^6$ | Actuator constant $T_1 = 0.0028$ |
| Time delay $t_d$ = 0.048 | |

| Quadratic function coefficients of fuel consumption against DC source: | |
|---|---|
| $c_0 = 2.2361,\ b_0 = 0.136,\ a_0 = 0.0007$ | $c_1 = 4.4645,\ b_1 = 0.1366,\ a_1 = 0.0003$ |

**Table A2.** DGRS parameters.

| Li-ion Battery: |
|---|
| Battery constant voltage $E_0 = 1200$ V |
| Internal resistance R $= 1 \times 10^{-4}$ Ω |
| Inverse of exponential zone time constant B = 0.31 (Ah)$^{-1}$ |
| Polarisation constant K = 0.026 Ω |
| Exponential zone amplitude A = 58.78 V |
| Initial state of charge $SOC_{initial}$ = 65% |

| DC to DC converter: |
|---|
| Capacitor $C_o = 1 \times 10^{-3}$ F    Capacitor $C_p = 1 \times 10^{-2}$ F |
| Capacitor $C_s = 1 \times 10^{-2}$ F |
| Inductance $L_s = 8 \times 10^{-7}$ H    Inductance $L_{dc} = 5 \times 10^{-5}$ H |
| Switching frequency $f_s$ = 20,000 Hz |
| Duty cycle D = 50% |

| DC-link capacitor: $C_t$ = 0.006 F |
|---|

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
