# Peer review of "An Experimental Investigation into the Feasibility of a DC Hybrid Power Plant for a Northern Sea Route Ship"

_jmse, doi:10.3390/jmse11091653_

Round 1
Reviewer 1 Report
1. The author’s manuscript said “35% fuel saving compared with the diesel electric configuration during a scaled-down NSR scenario.”
Please show the equation of the fuel saving.
2. The author’s manuscript said “To evaluate the performance of the hybrid power system with the proposed optimisation control strategy, lab-scale experiments have been conducted to compare the proposed system with a conventional hybrid system.”
Please show the equation of the performance of the hybrid power system.
3. The author’s manuscript said “The specific fuel consumption and various losses in the power sources were analysed to develop an efficiency-optimisation control strategy for the proposed DC hybrid power system.”
Please show the equation of an efficiency-optimisation control strategy for the proposed DC hybrid power system.
4. Please show the equation of the AVM rectifier.
5. The author’s manuscript said “DGRS will always be kept on at this stage to match the high loading power.”
Please show the equation of DGRS using the high loading power.
6. Please show the equation of the FC per hour(DC power).
7. The author’s manuscript said “An observation from the results of their work is that they present certain delays in the experimental signals when compared to the simulation results, particularly in the cases of the output power percentage of DGRS.
Please show the equation of the simulation results.
8. Please show the equation of Fuel consumption in Fig. 21.
9. The author’s manuscript said “In this paper, system components of a DC shipboard power system have been mathematically modelled and a 3-level control strategy has been developed based on the proposed hybrid DC power system.”
Please show the equation of the proposed hybrid DC power system.
10. The author’s manuscript said “An efficiency optimisation algorithm has been proposed rather than a rule-based Energy Management System to further consider the losses in the 598energy storage system.”
Please show the equation of the losses in the energy storage system
Please show the figure of the losses in the energy storage system
Author Response
Dear Reviewer 1,
Thank you for your constructive comments and please find our response in the attachment.
Yours sincerely,
Kayvan Pazouki

Reviewer 2 Report
some information is missing in the introduction to follow the consideration of the lab. details regarding consistency were added to make some changes for a better tracking of the manuscript. Quite interesting piece of work.

small paragraphs would be better to describe the simulations and the results, the figures need to be next to the descriptions for a better follow up.
Author Response
Dear Reviewer 2,
Thank you for your constructive comments and please find our response in the attachment.
Yours sincerely,
Kayvan Pazouki

Reviewer 3 Report
Please check the comments in the attached file.

Author Response
Dear Reviewer 3,
Thank you for your constructive comments and please find our response in the attachment.
Yours sincerely,
Kayvan Pazouki

Round 2
Reviewer 1 Report
1. It is good enough to publish the JMSE.
1. It is good enough to publish the JMSE.
Reviewer 3 Report
It seems that the authors put a lot of effort into correcting the paper.
Although corrections and clarifications have been added to many parts, there is no major change in the idea that experiments should be carried out based on theoretical analysis.
It is meaningful that the experiments was conducted to verify the effectiveness of hybrid propulsion system. However, it’ll be another problem to publish an experimental results different from the theoretical environment in the paper.
The topic is good and the contents are meaningful, but it is recommended to organize the results with the experimental results that match the theoretical analysis and the experimental environment as much as possible.
